# Achieving High-Strength and Toughness in a Mg-Gd-Y Alloy Using Multidirectional Impact Forging

**DOI:** 10.3390/ma15041508

**Published:** 2022-02-17

**Authors:** Songhe Lu, Di Wu, Ming Yan, Rongshi Chen

**Affiliations:** 1Academy for Advanced Interdisciplinary Studies, Southern University of Science and Technology, Shenzhen 518055, China; lush3@sustech.edu.cn (S.L.); yanm@sustech.edu.cn (M.Y.); 2Shi-Changxu Innovation Center for Advanced Materials, Institute of Metal Research, Chinese Academy of Sciences, 72 Wenhua Road, Shenyang 110016, China; dwu@imr.ac.cn

**Keywords:** grain refinement, high strength and toughness, {10–12} twin, Mg-Gd-Y alloy, multidirectional impact forging

## Abstract

High strength and toughness are achieved in the Mg-4.96Gd-2.44Y-0.43Zr alloy by multidirectional impact forging (MDIF). The forged sample has a fine-grained microstructure with an average grain size of ~5.7 µm and a weak non-basal texture, and it was characterized by an optical microscope (OM), scanning electron microscope (SEM), and electron back-scattering diffraction (EBSD). Tensile results exhibit the tensile yield strength (TYS) and static toughness (ST) of as-homogenized alloy dramatically increased after forging and aging, i.e., the TYS increased from $135_{-5}^{+4}$ MPa to $337_{-2}^{+2}$ MPa, and the ST enhanced from 22.$0_{-0.5}^{+0.3}$ MJ/m^3^ to 50.$4_{-5.4}^{+5.3}$ MJ/m^3^. Specifically, the forged Mg-Gd-Y-Zr alloy owns higher TYS than that of commercial rolled WE54 (Mg-5.25Y-3.5Nd-0.5Zr) and WE43 (Mg-4.0Y-3.0Nd-0.5Zr) alloys.

## 1. Introduction

Mg and its alloys, as structural materials for automobile and electronics industries, can meet the demands of weight reduction and increasing vehicle efficiency [1,2]. Unluckily, they exhibited relatively low yield strength and toughness in comparison to their competitors Al and Ti alloys [3]. To improve their mechanical property, some strengthening mechanisms (including solution and precipitation strengthening) have been applied to block the dislocation glide and achieve the enhancement of the yield strength but the degradation of the toughness [4,5,6]. In contrast, fine-grained strengthening has been considered as a promising way for synchronously enhancing the strength and toughness of Mg alloys. Furthermore, the corresponding strengthening effect in Mg alloys is more remarkable due to their larger slope coefficient of the Hall-Petch relationship (~3 times larger than that of aluminum alloy) [7].

Fine-grained AZ31 [8], AZ61 [9], AZ80 alloys [10] have been produced mainly by conventional deformation routes including rolling, extrusion, and forging, and the corresponding recrystallization predominantly proceeded from prior grain boundaries and then consumed the entire deformed microstructure easily. However, the above recrystallization mechanism was seriously suppressed in some RE (rare earth)-containing Mg alloys [11,12]. For instance, the microstructure of Mg-8.2Gd-3.8Y-1.0Zn-0.4Zr (wt.%) alloy rolled at 400 °C consisted of a large volume of residual coarse grains and some recrystallization distributed at the prior grain boundary [11]. A similar recrystallization behavior also was witnessed in the microstructure of extruded Mg-7.5Gd-2.5Y-3.5Zn-0.9Ca-0.4Zr (wt.%) alloy [12]. This should be originated from the strong solute drag or pinning effect of RE-rich precipitates on dislocation glide and rearrangement, especially for those grain boundaries having intensive segregation of solute atoms. In other words, these grain boundaries have already lost the function of recrystallization nucleation. To produce the fine-grained RE-containing Mg alloys, it is necessary to offer appropriate nucleation sites in the grain interior for recrystallization.

Mechanical twinning can serve as an important deformation mechanism in the plastic deformation of the Mg-RE alloys [13]. The corresponding twin boundaries generated in the grain interior may be the preliminary candidate for recrystallization nucleation [14]. For common {10–12} extension, {10–11} contraction and {10–11}–{10–12} double twins, the latter two is difficulty to be utilized well due to their larger critical resolved shear stress (CRSS ≥ 70 MPa) [15,16] and crack tendency [17]. In contrast, the former {10–12} twin with relatively low CRSS (~10 MPa) is easy to be activated and accommodates the extension strain along the c-axis [18]. Hence, the {10–12} twin boundary may take the place of the initial grain boundary and act as preference nucleation sites for recrystallization of the RE-containing Mg alloys.

Recently, {10–12} twin activated by multidirectional impact forging (MDIF) has been successfully applied to activate {10–12} twin and produce fine-grained Mg-6.68Gd-5.9Y-0.48Zr (wt.%) alloy [19,20]. In contrast to recrystallization assisted by dislocation rearrangement, {10–12} twin induced recrystallization has many advantages as follows. Firstly, mutual intersections of two-dimensional {10–12} twin boundaries are easy to form three-dimensional recrystallization nuclei. Secondly, the fresh twin boundary can store enough strain energy to promote recrystallization nucleation through the repeated twin-dislocation interaction during the dynamic forging procedure [19]. Finally, recrystallization grains initiated from {10–12} twin boundary are more effective in achieving microstructure refinement of the Mg-Gd-Y alloy alloys.

In the present work, a high-strength and toughness Mg-Gd-Y alloy has been developed by MDIF using {10–12} twin and correlated recrystallization. Then, the contribution of {10–12} twins to microstructure refinement and property enhancement of Mg-Gd-Y-Zr alloy have been discussed in detail.

## 2. Materials and Methods

The material used in the present study was GW52 alloy ingot (Mg-4.96Gd-2.44Y-0.43Zr, mass fraction, %). The as-cast GW52 alloy ingot was homogenized at 480 °C for 8 h with subsequent cooling in air, and then it was cut into some cubic sample with dimensions of 70 × 70 × 70 mm^3^. These cubic samples were first heated to 480 °C in an electric resistance furnace and kept for 60 min. In addition, the heating rate is 15 °C/min. Then, the multidirectional impact forging (MDIF) was carried out using an industrial air pneumatic hammer machine (Shandong Chu Hang Heavy Industry Machinery Co., Ltd., Weifang, China) with a load gravity of 250 Kg. A pass strain ~0.05 was applied and the average strain rate was around 20 s^−1^. The procedure during the MDIF was shown in Figure 1a. One of the cubic samples was forged to 200 passes with an intermediate annealing treatment for ~10 min at 480 °C, and it was termed as MDIF100+100 sample (symbol + means an intermediate annealing treatment for ~10 min after the first 100 forging passes). The entire forging and annealing process finished in ~15 min. After carefully checking, the MDIF100+100 sample was free from any surface defects, as shown in its macro-morphology image Figure 1b.

To examine the microstructure and mechanical property, the MDIF100+100 sample was sectioned in the center along the last forging direction (LFD), as shown in Figure 1c. The corresponding specimens for optical microscopy (OM) (Carl Zeiss AG, Oberkochen, Germany), scanning electron microscopy (SEM) (Carl Zeiss AG, Oberkochen, Germany), and tension test was machined from the center part, as indicated by red arrows in Figure 1c. The electron back-scattering diffraction (EBSD) observations were carried out using a TESCAN-MIRA3 SEM (TESCAN, Brno, Czech Republic) operating at 20 kV and applying a corresponding probe current of 60 nA. Orientation imaging microscopy was measured at a step size of 1 μm and the acquired EBSD data was analyzed using the software of Aztec Crystal 4.3 (Oxford Instruments, Abingdon, UK). Texture analysis was conducted using the Schultz reflection method through X-ray diffraction and calculated pole figures were obtained using the DIFFRAC plus TEXEVAI software (Bruker, Karlsruhe, Germany). The tensile specimens in dog bone shape were cut from the central part of the forged cubic sample with a gauge length of 20 mm, a width of 3 mm, and a thickness of 2.5 mm. The compression specimens in-cylinder were also machined from the same position with a height of 15 mm and a diameter of 10 mm, as shown in Figure 1c.

## 3. Results

### 3.1. Initial State of the Alloy

Figure 2 displays the EBSD results of the microstructure of GW52 alloy after homogenization at 480 °C for 8 h. The inverse pole figure exhibit many equiaxed grains with random orientation in the microstructure of as-homogenized alloy, as shown in Figure 2a. This is consistent with the {0001} pole figure in Figure 2d. The corresponding grain boundary map consists of an as-homogenized microstructure with some second phases in Figure 2b. Most of these phases in Mg-Gd-Y were identified as cuboid-shaped phases in previous reference [20], which were detected as YH_2_ through a combined analysis of secondary ion mass spectrometry (SIMS) and X-ray tomography (XRT). Many black dots in Figure 2b may be the corrosion products of the second phase produced by electrolytic polishing. The average grain size of 47 μm has been evaluated through the grain size distribution map in Figure 2c.

### 3.2. Microstructure and Mechanical Property of MDIF100+100 Sample

Figure 3 gives OM and corresponding {0001} macro texture of the MDIF100+100 sample of GW52 alloy. It can be observed a deformed microstructure in low magnification of OM in Figure 3a. High-magnification observation captured some coarse grains ~10 µm and a large number of small grains ~1 µm, as indicated by their respective arrows in Figure 3b. Besides, the corresponding {0001} macrotexture in Figure 3c illustrates a non-basal texture with many peaks (the intensity ≤ 3.79 multiples of random (mrd)).

The corresponding EBSD results of the final forging microstructure were shown in Figure 4a–e. We noted that flourishing twins were activated in the microstructure, especially in several coarse regions, as shown in Figure 4a,b. The special boundaries results in Figure 4b demonstrate that the fraction of {10–12} twin boundary (<11–20> 86°)is 11.8%, while the fraction of {10–11} twin boundary (<11–20> 56°) and {10–11}–{10–12} twin boundary <11–20> 38° are 0.23% and 0.40%, respectively. This is consistent with the larger fraction of {10–12} twin boundaries in the misorientation angle of grain boundaries in Figure 4e. In addition, low angle grain boundaries (LAGBs) also have a larger number fraction in this forged alloy in Figure 4e. As shown in the grain size distribution histogram of Figure 4d, the final average grain diameter of the GW52 alloy was decreased to ~5.7 μm. Moreover, the fraction of the fine grins (≤20 μm) in the final forging microstructure was about 87%. In particular, the {0001} pole figure in Figure 4d exhibited the fine-grained alloy possess a non-basal texture (the intensity ≤ 4.13 mrd). Specifically, MDIF with intermediate annealing treatment is an effective route to produce fine-grained Mg-RE alloys with non-basal texture.

Figure 5 presents room temperature tensile engineering stress-strain curves of as-homogenized, forged, and aged GW52 alloy. The tension results were listed in Table 1. The static toughness (ST) usually can be measured by the true stress-strain curve, which can be obtained from their respective engineering stress-strain curve through data processing. The true stress can be calculated by this equation: S = σ* (1 + ε). Then, ST was measured by the equation ST = $\int_{0}^{\varepsilon_{k}}Sd\varepsilon$ [21,22]. In addition, S means true stress and εk is the value of true strain after the sample fractured. The corresponding results have been illustrated in Table 1.

It can be observed in Figure 5a that the tensile yield strength (TYS), uniform elongation (EL), and static toughness (ST) of as-homogenized GW52 alloy were $135_{-5}^{+4}$ MPa, 10.$0_{-0.4}^{+0.4}$%, and 22.$0_{-0.5}^{+0.3}$ MJ/m^3^, respectively. After the MDIF of 100+100 passes, the strength, ductility, and toughness of GW52 alloy were synchronously enhanced in Figure 5b, i.e., the TYS, EL, and ST dramatically increased to $286_{-3}^{+3}$ MPa, 13.$2_{-0.9}^{+0.8}$%, and 49.$9_{-1.7}^{+1.6}$ MJ/m^3^, respectively. It has been reported that aging treatment at 200 °C was usually applied to Mg-RE alloys to further enhance their strength [23,24,25]. In the present work, some tensile specimens of GW52 alloy were aged at 200 °C for different times (30 h, 60 h, and 120 h). The strength and toughness often were synchronously enhanced in Figure 5c–e, i.e., foraged alloy at 200 °C for 60 h, the TYS, UTS, and ST sharply increased to $337_{-2}^{+2}$ MPa, $361_{-1}^{+0}$ MPa, and 40.$1_{-5.3}^{+4.7}$ MJ/m^3^, respectively.

Figure 6a shows the comparison of mechanical properties in numerous forged Mg-Al-Zn alloys and Mg-Gd-Y-based alloys [10,26,27,28,29,30,31,32,33,34,35,36,37,38,39]. The forged Mg-Al-Zn-based alloys were generally located at the bottom corner [10,27,28,29,30,31,32,33], while the MDIF100+100 sample of GW52 alloy was located at the center part exhibiting a higher TYS. Furthermore, the forged GW52 alloy exhibited excellent strength–ductility balance in comparison to the forged Mg-Gd-Y-based alloys [34,35,36,37,38,39]. Specifically, the TYS and EL of the forged GW52 alloy are larger than that of rolling or extrusion of commercial WE54 and WE43 alloys produced by Magnesium Elektron Ltd. (Magnesium Elektron, Manchester, UK), in Appendix A, as shown in Figure 6a. Above all, the forged GW52 alloy demonstrates tension-compression yield symmetry along N3 forging direction (the ratio of CYS/TYS was ≈ 1.0) in Figure 6b, which means that we obtained yield isotropy Mg-Gd-Y alloy with high strength and toughness. This tension-compression yield symmetry character is rarely examined in the rolling sheet or extrusion of high-strength Mg-RE alloy [40]. It can be concluded that a fine-grained GW52 alloy has been produced by MDIF using {10–12} twin and correlated recrystallization with the excellent mechanical property. Thus, the contribution of {10–12} twins to microstructure refinement and property enhancement of GW52 alloy should be discussed in detail.

## 4. Discussion

### 4.1. Contribution of {10–12} Twins to Microstructure Refinement

It has been reported that {10–12} twins usually exerted different functions at dynamic forging and annealing treatment in our previous works [19,20]. At the dynamic forging process, MDIF with a high strain rate and small strain is favorable for twin nucleation but twin growth [20]. Additionally, the three directional loadings are beneficial for activating {10–12} twins in 79.8% of grains in the microstructure of the as-homogenized Mg-Gd-Y alloy [20]. Meanwhile, these twins can divide their parent grains, interact repeatedly with various dislocations, and then store enough strain energy, eventually promoting the nucleation of DRX [20]. With the progress of intermediate annealing, above {10–12} twins with high strain energy can act as preference nucleation sites for static recrystallization, thereby making for a preliminary microstructure refinement [19]. Next, this fine-grained microstructure was ready for the next generation of {10–12} twin. At the sequent dynamic forging passes, the fresh {10–12} twins can be activated again and subdivide those new grains achieving further microstructure refinement. It can be concluded that MDIF with intermediate annealing can effectively refine the microstructure of Mg-Gd-Y alloy through {10–12} twin and correlated recrystallization.

The above microstructural results can be evidenced by the fine-grained microstructure with extensive {10–12} twins, as shown in inverse pole figure map and band contrast map of Figure 4a,b. The distribution histogram of grain size had been examined in Figure 4d, and the average grain size of the GW52 alloy sharply decreased from ~47 µm of as-homogenized alloy to ~5.7 µm. We noted that the fraction of the fine grins (≤20 µm) dramatically increased to 87%. This means that MDIF with intermediate annealing had produced a fine-grained Mg-Gd-Y alloy using {10–12} twin and correlated recrystallization.

### 4.2. High Strength and Toughness GW52 Alloy

After the MDIF of 100+100 forging passes, the TYS of GW52 alloy increased from $135_{-5}^{+4}$ MPa to $286_{-3}^{+3}$ MPa and the ST also increased from 22.$0_{-0.5}^{+0.3}$ MJ/m^3^, to 49.$9_{-1.7}^{+1.6}$ MJ/m^3^. As a result, the MDIF realized 151 MPa increment (112%) for TYS and 22.0 MJ/m^3^ increment (117%) for ST. Meanwhile, the average grain size of the alloy sharply decreased from ~47 µm of as-homogenized alloy to ~5.7 µm. The strengthening and toughening mechanisms will be discussed in detail as follows.

For the strengthening mechanism, fine-grained strengthening and twin strengthening should be focused on in the present work. Based on the stress intensity factor, k, obtained using the Hall–Petch relationship of GW53 alloy [41], the increment of TYS induced by grain refinement was calculated at about 112 MPa. It can be concluded that fine-grained strengthening determines the increment of TYS of GW52 alloy. Besides, flourishing twins were activated in the microstructure, especially in several coarse regions, as shown in Figure 4a,b. The special boundaries results in Figure 4b demonstrate that the fraction of {10–12} twin boundary (<11–20> 86°) is 11.8%, while the fraction of {10–11} twin boundary (<11–20> 56°) and {10–11}–{10–12} twin boundary <11–20> 38° are 0.23% and 0.40%, respectively. As a result, the crack tendency of {10–11} compression and {10–11}–{10–12} double twin can almost be neglected [42,43]. As a result, a large number of {10–12} twin boundaries as typical two-dimensional planar defects can interact with basal slip and even impede the dislocation movement [44]. Therefore, fine grain and twin strengthening play a key role in the high-strength GW52 alloy.

For the toughening mechanism, intergranular deformation including grain sliding and grain rotation usually serves as the dominating deformation mechanism in the plastic deformation of fine-grained GW52 alloy (~5.7 µm). Meanwhile, deformation mechanisms will have a significant transition from primary basal slip to some potential non-basal slip in fine-grained alloy with the grain size < 10 µm [45]. In addition, a large volume of {10–12} twin boundaries as typical two-dimensional planar defects exerted an obstructing effect on dislocation glide and allow the penetration of dislocation under the high stress-concentration [44]. On the other hand, the fine-grained GW52 alloy with a random texture tends to activate various dislocation slip, which was evidenced by their higher average SF (Schmid factor) value (≥0.28) for various slip systems (basal slip <a>: {0001} <11–20>, prismatic slip <a>: {1–100} <11–20>, pyramidal slip <a>: {1–101} <11–20>, and pyramidal slip <c+a>: {11–22} <11-2-3>), as shown in Figure 7. In other words, various slips will be activated and steadily accommodate the tensile strain according to the Von Mises criteria [46]. All of these changes can ensure a uniform deformation and suppress the premature failure of Mg alloy. Hence, the fine-grained microstructure obtained by MDIF plays an important role in this high strength and toughness Mg-Gd-Y alloy.

## 5. Conclusions

Multidirectional impact forging has been applied to grain refinement and property enhancement of Mg-Gd-Y alloy in the present work. The main results can be summarized as follows:(1)Multidirectional impact forging has been proved to be an efficient methodology in grain refinement and property improvement of Mg-Gd-Y alloy. After the MDIF of 100+100 forging passes, the TYS of GW52 alloy increased from $135_{-5}^{+4}$ MPa to $286_{-3}^{+3}$ MPa and the ST also increased from 22.$0_{-0.5}^{+0.3}$ MJ/m^3^, to 49.$9_{-1.7}^{+1.6}$ MJ/m^3^. As a result, the MDIF realized 151 MPa increment (112%) for TYS and 22.0 MJ/m^3^ increment (117%) for ST simultaneously.(2)The MDIF100+100 sample of GW52 alloy has a relatively fine-grained microstructure ~5.7 µm, exhibiting a random texture. Furthermore, high TYS of $337_{-2}^{+2}$ MPa, EL of 11.$5_{-1.3}^{+1.3}$%, and ST of 50.$4_{-5.4}^{+5.3}$ MJ/m^3^ were gained in the GW52 alloy developed by MDIF and aging treatment.(3)The forged GW52 alloy exhibits yield isotropy (the ratio of compression yield strength/tensile yield strength along the forging direction is ≈1.0). Compared with forged Mg-Al-Zn and Mg-Gd-Y based alloys, the forged GW52 alloy exhibited excellent strength–ductility balance resulting from the combined effect of fine-grained strengthening, twin strengthening, and its non-basal texture.

## Figures and Tables

**Figure 1 materials-15-01508-f001:**
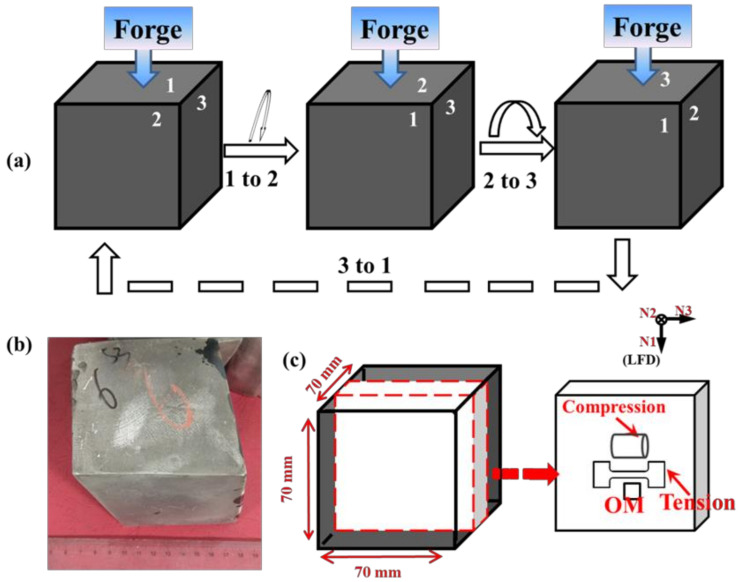
(**a**) Schematic illustrations of the MDIF process, (**b**) macro-morphology of MDIF100+100 sample and (**c**) schematic diagram describing the orientations of the tensile specimens and the compression specimens relative to the forged billet.

**Figure 2 materials-15-01508-f002:**
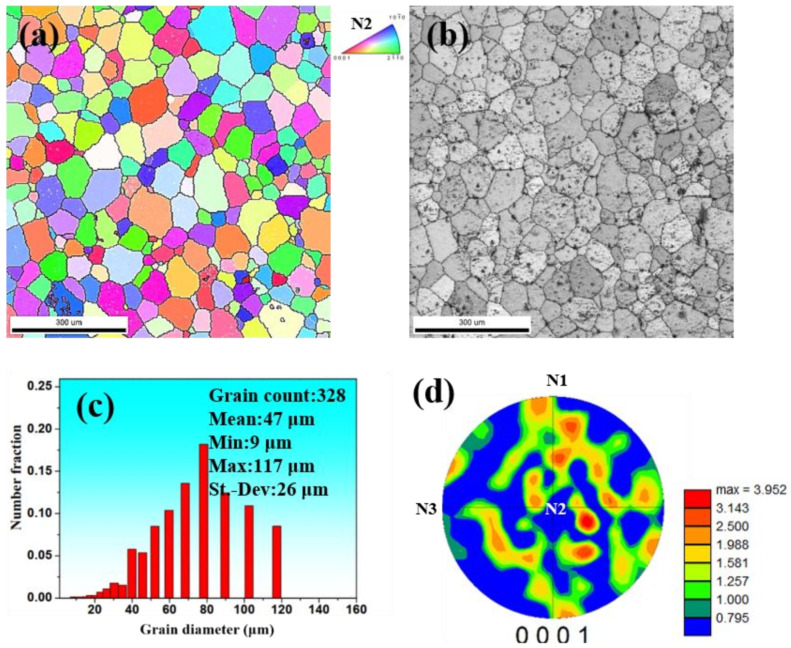
(**a**) Inverse pole figure, (**b**) grain boundary image, (**c**) grain size distribution map, and (**d**) {0001} pole figure of as-homogenized GW52 alloy.

**Figure 3 materials-15-01508-f003:**
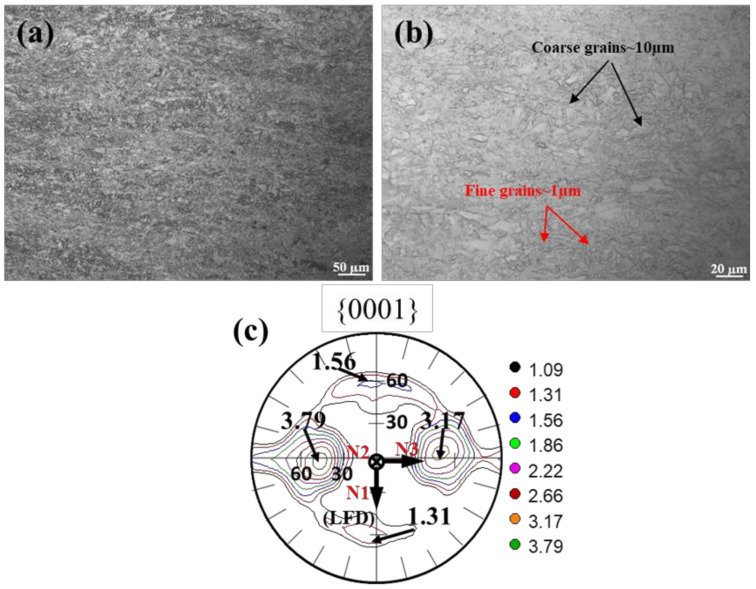
(**a**) Low magnification and (**b**) high magnification of OM as well as corresponding (**c**) {0001} macrotexture of MDIF100+100 sample of GW52 Mg alloy at the center location (~35 mm).

**Figure 4 materials-15-01508-f004:**
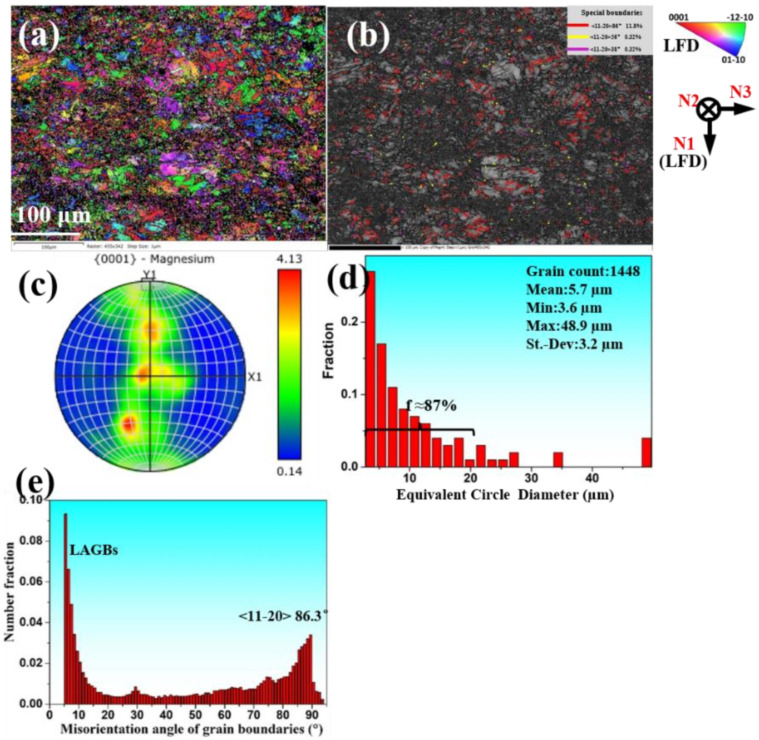
(**a**) Inverse pole figure map, (**b**) band contrast map, (**c**) {0001} pole figure, (**d**) grain size distribution histogram, and (**e**) misorientation angle of grain boundaries of MDIF100+100 sample of GW52 alloy at the center location (~35 mm).

**Figure 5 materials-15-01508-f005:**
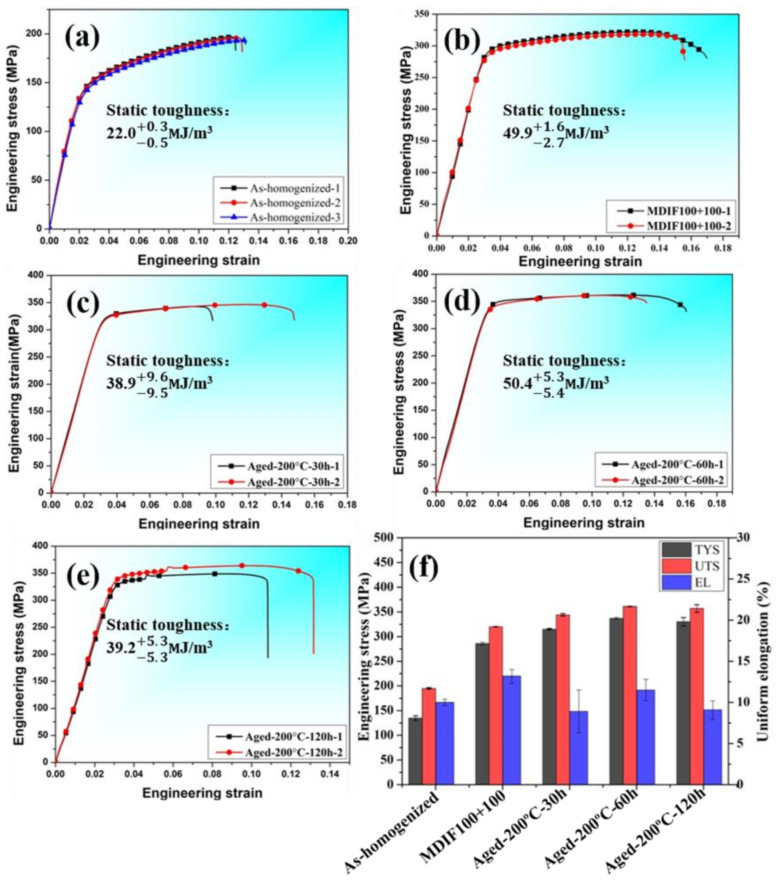
Room temperature tensile engineering stress-strain curves (**a**–**e**) and corresponding properties (**f**) of as-homogenized, MDIF100+100, and aged MDIF100+100 samples of GW52 alloy with their respective static toughness: (**a**) as-homogenized alloy, (**b**) MDIF100+100, (**c**) aged at 200 °C for 30 h, (**d**) aged at 200 °C for 60 h, (**e**) aged at 200 °C for 120 h.

**Figure 6 materials-15-01508-f006:**
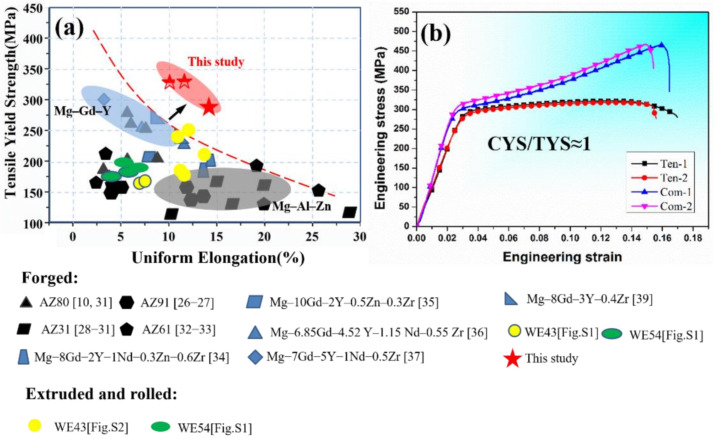
(**a**) Relationships between tensile yield strength and elongation of forged Mg-Al-Zn based and Mg-Gd-Y-based alloys as well as (**b**) tension-compression engineering stress-strain curves of MDIF100+100 sample of GW52 alloy.

**Figure 7 materials-15-01508-f007:**
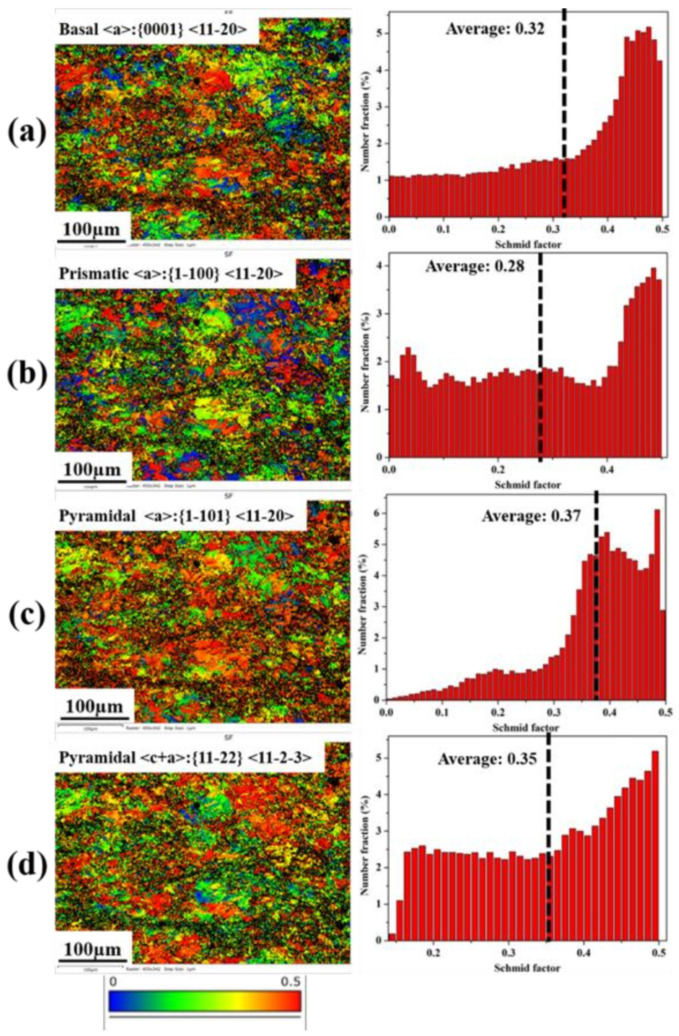
SF maps and corresponding distribution histograms for various slip systems of MDIF100+100 sample of GW52 alloy: (**a**) basal slip <a>:{0001} <11–20>, (**b**) prismatic slip <a>:{1–100} <11–20>, (**c**) pyramidal slip <a>:{1–101} <11–20>, (**d**) pyramidal slip <c+a>:{11–22} <11-2-3>.

**Table 1 materials-15-01508-t001:** Room temperature tensile and compressive properties of GW52 alloy.

	TYS (MPa)	UT/CS(MPa)	EL (%)	ST(MJ/m^3^)	CYS/TYS
As-homogenized-1	139	197	9.6	21.5	
As-homogenized-2	137	196	10.0	22.2	
As-homogenized-3	130	194	10.4	22.3	
MDIF100+100-ten-1	289	321	14.0	52.5	≈1
MDIF100+100-ten-2	283	319	12.3	47.2	
MDIF100+100-com-1	294	465	13.0	-	
MDIF100+100-com-2	310	469	11.0	-	
Aged-200 °C-30 h-1	316	341	6.2	29.3	
Aged-200 °C-30 h-2	313	346	11.5	48.4	
Aged-200 °C-60 h-1	339	361	12.8	55.7	
Aged-200 °C-60 h-2	335	360	10.2	45.0	
Aged-200 °C-120 h-1	322	364	7.9	33.9	
Aged-200 °C-120 h-2	339	349	10.2	44.5	

TYS, tension yield strength; UTS, ultimate tensile strength; EL, uniform elongation; CYS, compression yield strength; ST, static toughness.

## Data Availability

The data presented in this study are available upon request from the corresponding author. The data are not publicly available due to the requirements of related projects.

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
