# Peer review of "Achieving High-Strength and Toughness in a Mg-Gd-Y Alloy Using Multidirectional Impact Forging"

_materials, 2022, doi:10.3390/ma15041508_

Round 1

Reviewer 1 Report

The work presented in this paper can be suitable for Materials Journal and covers some new aspects of the high-strength and toughness in Mg-Gd-Y alloy through multidirectional impact forging. However, reviewer would like to make the following comments

Introduction

Mention the chemical composition of   WE54 and WE43

Research methodology

What is the heating rate for 480 C?

How did the authors calculate the average strain?

Results and discussion

Effect of multidirectional impact forging on GW52 (Mg-4.96Gd-2.44Y-76 0.43Zr) was studied in this manuscript. The intermetallic phase (YH2) was also detected at Fig.2. Thus, the authors have to justify how the phase was formed while hydrogen not included in the chemical composition  

Author Response

Please write down "Please see the attachment." in the box if you only upload an attachment.

Reviewer 2 Report

The study "Achieving high-strength and toughness in a Mg-Gd-Y alloy using multidirectional impact forging" is an interesting peace of work devoted to a promisssing methode of multidirectional forging. The paper corresponds well to the journal scope and can be published after minor revision. 

Comments

In Fig. 7. distribution is shown. I recommend drawing a middle line, this way it's easier to estimate the size.
In Fig. 4b no red/green lines are visible, the quality of the drawing needs to be improved.

Author Response

(The authors gave the same response as above.)

Reviewer 3 Report

The article "Achieving high-strength and toughness in a Mg-Gd-Y alloy using multidirectional impact forging" is dedicated to the Mg-Gd-Y-Zr alloy with high mechanical properties. The microstructure was characterized by an optical microscope and scanning electron microscope, and electron back-scattering diffraction. The experimental results are reliable and the paper could be published in Materials journal with Minor corrections:

Fig. 2. shows the microstructure on which the EBSD analysis was done. It seems that the microstructure is all in pitting corrosion. Why are black dots visible on the microstructure? Either explain why the sample is oxidized and corroded or do not include this figure in the article at all.

In Figure 2a, the caption “N2 direction” is not good. It is necessary to move the caption to another place for a visible figure.
